# “K-Powder” Exposure during Adolescence Elicits Psychiatric Disturbances Associated with Oxidative Stress in Female Rats

**DOI:** 10.3390/ph15111373

**Published:** 2022-11-09

**Authors:** Sabrina de Carvalho Cartágenes, Cinthia Cristina Sousa de Menezes da Silveira, Bruno Gonçalves Pinheiro, Luanna Melo Pereira Fernandes, Sarah Viana Farias, Natália Harumi Correa Kobayashi, Pablo Henrique Franco Santos de Souza, Alejandro Ferraz do Prado, Maria Karolina Martins Ferreira, Rafael Rodrigues Lima, Edivaldo Herculano Correa de Oliveira, Francisco Canindé Ferreira de Luna, Rommel Mário Rodríguez Burbano, Enéas Andrade Fontes-Júnior, Cristiane do Socorro Ferraz Maia

**Affiliations:** 1Laboratory of Pharmacology of Inflammation and Behavior, Health Sciences Institute, Pharmacy College, Federal University of Pará, Belém 66075-900, PA, Brazil; 2Physiological and Morphological Sciences Department, Biological and Health Science Centre, State University of Pará, Belém 66087-662, PA, Brazil; 3Laboratory of Pharmacology and Toxicology of Cardiovascular System, Institute of Biological Science, Federal University of Pará, Belém 66075-900, PA, Brazil; 4Laboratory of Functional and Structural Biology, Institute of Biological Sciences, Federal University of Pará, Belém 66075-900, PA, Brazil; 5Laboratory of Cytogenomics and Environmental Mutagenesis, Environment Section (SAMAM), Evandro Chagas Institute (IEC), Ananindeua 67030-000, PA, Brazil; 6Laboratory of Molecular Biology, Ophir Loyola Hospital, Belém 66063-240, PA, Brazil

**Keywords:** ketamine, withdrawal, adolescent, anxiety, depression, oxidative stress

## Abstract

Ketamine, also called ‘K-powder’ by abusers, an analog of phencyclidine, primarily acts as an antagonist of N-methyl-D-aspartic acid (NMDA) receptors, therapeutically used as an anesthetic agent. Ketamine also stimulates the limbic system, inducing hallucinations and dissociative effects. At sub-anesthetic doses, ketamine also displays hallucinatory and dissociative properties, but not loss of consciousness. These behavioral consequences have elicited its recreational use worldwide, mainly at rave parties. Ketamine is generally a drug of choice among teenagers and young adults; however, the harmful consequences of its recreational use on adolescent central nervous systems are poorly explored. Thus, the aim of the present study was to characterize the behavioral and biochemical consequences induced by one binge-like cycle of ketamine during the early withdrawal period in adolescent female rats. Adolescent female Wistar rats (*n* = 20) received intraperitoneally administered ketamine (10 mg/kg/day) for 3 consecutive days. Twenty-four hours after the last administration of ketamine, animals were submitted to behavioral tests in an open field, elevated plus-maze, and forced swimming test. Then, animals were intranasally anesthetized with 2% isoflurane and euthanized to collect prefrontal cortex and hippocampus to assess lipid peroxidation, antioxidant capacity against peroxyl radicals, reactive oxygen species, reduced glutathione, and brain-derived neurotrophic factor (BDNF) levels. Our results found that 24 h after recreational ketamine use, emotional behavior disabilities, such as anxiety- and depression-like profiles, were detected. In addition, spontaneous ambulation was reduced. These negative behavioral phenotypes were associated with evidence of oxidative stress on the prefrontal cortex and hippocampus.

## 1. Introduction

K-powder, as called by ketamine addicts, belongs to the arylcyclohexylamine chemical class, and was synthesized in 1962 by Calvin Stevens [1,2,3,4]. In humans, the first report of ketamine drug addiction consisted of 20 volunteer prisoners in 1964, which also established ketamine as an analgesic and anesthetic agent [5]. Henceforward, additional pharmacological effects were described, such as effects on pediatric pain, antidepressant action, and anti-suicide properties [6,7,8,9,10,11]. However, schizophrenia-like psychotic symptoms were reported [12].

Despite the relevant clinical findings, the psychedelic effects elicited by ketamine attract the attention of addicts for new drugs, which characterizes the novel psychoactive substances (NPS) [13]. In the recreational context, ketamine is popularly known as “Special K”, “Vitamin K”, “Valium Cat”, “K” or “Kate” [14,15], and addicts acquire the substance in liquid form through clandestine laboratories or veterinary clinics, evaporate the liquid to produce a powder to administer by the intranasal or oral route (K-powder) [2,15]. Recently, the report of the United Nations Office on Drug Control—UNODC (2021) demonstrated that in the last 15 years, there has been a dominance in ketamine seizures when compared to other drugs with hallucinogenic properties, of which adolescents and female groups exhibited higher increments among consumers [16].

Acute consumption of ketamine elicits hallucinations, time and space distortions, oneiric experiences, or mood swings [13,15,17]. In addition, “fusion in the environment” or “outside of the body” experiences (sometimes referred to as K-hole) associated with profound fear and anxiety were reported [2,12,14,17,18]. Post-withdrawal residual consequences were reported by Morgan and colleagues, for which long-term recreational use of ketamine promotes persistent neuropsychiatric symptoms, usually characterized as schizophrenia-like features, as well as impairment of working and episodic memories, and semantic processing [19,20]. However, scarce studies have exploited ketamine exposure during the adolescence period.

In adolescence, 10–19 years in humans is equivalent to 28–42 postnatal days in rodents [21,22,23], and the brain undergoes refinement processes, such as synaptic pruning, which facilitates synaptic circuitry improvement to establish an adequate synaptic network in several complex brain functions [21,24]. Thus, psychotropic substances may interfere with the normal development of the adolescent brain, reflected in behavioral disruption [24].

Morgan and Curran [18] attribute the overactivity of NMDA glutamate receptors in the cerebral cortex and hippocampus to ketamine neurotoxicity, as well as disruption of modulatory monoamines transmission, such as dopamine and serotonin (5-HT) on the striatum and cortex [18]. The blockade of continuous NMDA receptors by ketamine elicits cell death in the developing brain through a mechanism that involves a compensatory regulation of subunits of the NMDA receptor, which consequently induces intracellular toxic accumulation of calcium, oxidative stress, and activation of the nuclear factor kappa B (NF-kB) signaling pathway [25]. These toxicological events increase the vulnerability of neurons, even following ketamine withdrawal [26,27].

In fact, our group has reported that immediate withdrawal of subanesthetic doses of ketamine in adolescent rats increased the levels of malonaldehyde (MDA) and nitrite (NO^2−^) in the blood and hippocampus [28]. This fact is possible, due to the vulnerability of the adolescent central nervous system (CNS), in which important structural and functional changes of synaptic plasticity and neural connectivity occur during CNS development and maturation [24,29]. In this context, the present work claims to explore the emotional behavior, biochemical analyses, and BDNF levels on the prefrontal cortex and hippocampus of adolescent female rats submitted to a ketamine binge-like cycle.

## 2. Materials and Methods

### 2.1. Animals and Ethical Aspects

Twenty female Wistar rats (21 postnatal days) were obtained from Animal Facility of Federal University of Pará (UFPA) and kept in collective cages (5 animals per cage) in an experimental room under standard temperature conditions (25 ± 1 °C), exhaustion, 12 h light/dark cycle (6 a.m. to 6 p.m.), water and food ad libitum. This study was approved by the Ethics Committee on the Use of Animals of UFPA, under license CEUA number 9692050417, following criteria and standards recommended by the National Council for the Control of Animal Experimentation (CONCEA/Brazil) and NIH Guide for the Care and Use of Laboratory Animals. In the behavioral experimental rooms, fluorescent lights (12 lux), temperature (25 ± 1 °C), and noise attenuation were employed.

### 2.2. Experimental Design

Animals were randomly divided into two groups (10 animals/group). According to neurodevelopmental studies, to assess the initial stage of adolescence, from 28 to 30 days old, animals received saline or dextroketamine (Cristália Produtos Químicos e Farmacêuticos Ltd., São Paulo, Brazil) intraperitoneally (i.p.) 10 mg/kg/day; final volume of 0.1 mL/100 g of body weight) for three consecutive days [28,30,31]. Dextroketamine was elected because of its higher pharmacological activity than R(-)-ketamine enantiomer or racemic mixture [32,33]. On the 31st postnatal day, animals were submitted sequentially to open field, elevated plus-maze, and forced swim tests, followed by biological sample collection (Figure 1). Following behavioral tests, animals were anesthetized with isoflurane 2% intranasally and euthanized by decapitation. Biological samples (i.e., hippocampus and prefrontal cortex) were dissected and divided for oxidative stress evaluation (*n* = 5 animals/group) and reactive species oxygen (ROS) plus brain-derived neurotrophic factor (BDNF) levels (*n* = 3–5 animals/group).

### 2.3. Behavioral Assays

All behavioral assays were performed at the Laboratory of Pharmacology of Inflammation and Behavior (LAFICO/UFPA) from 7 a.m. until 6 p.m. to avoid animal’s circadian cycle interference. Behavioral trials were performed twenty-four hours after the last administration of ketamine. On behavioral test day, animals were conducted to experimental room and acclimated for 1 h. In addition, animals were randomly submitted to behavioral tests, as well as the double-blind requirement was applied.

#### 2.3.1. Open Field

Open field device consists of an acrylic black square (100 × 100 × 40 cm) virtually divided into 25 quadrants (20 × 20 cm), identifying two zones, central area (9 quadrants) and periphery (16 quadrants). To evaluate spontaneous locomotor activity, two behavioral parameters were assessed, i.e., total distance traveled and number of rearing, for 5 min, which reflects horizontal and vertical exploitation [34]. In addition, to measure anxiety-like features, spent time and distance traveled in the center of apparatus were recorded [35,36]. All movements throughout the session were recorded using a video camera positioned above the arena, for further analysis by ANY-maze software (San Diego, CA, USA).

#### 2.3.2. Elevated Plus-Maze

Elevated plus-maze is a cross-shaped wooden equipment raised 50 cm from the floor, composed of two closed arms (50 × 10 × 40 cm) and two opposite open arms (50 × 10 × 1 cm), employed to validate responses related to anxiety in rodents based on the spontaneous exploratory activity, which reflects the conflict between natural tendency to explore new environments and to avoid aversive situations [35,37]. Briefly, animals were individually positioned on the center of the apparatus facing a closed arm. Spontaneous ambulation was allowed for 5 min. Enclosed arms entries (EAE) were admitted as motor validation. Percentual of open arms entries (%OAE) and open arms time (%OAT) were calculated according to the formulas [(OAE/OAE + EAE) × 100; (OAT/OAT + EAT) × 100] and adopted as anxiety-like parameters [34,38,39].

#### 2.3.3. Forced Swimming Test

Forced swimming device consists of an acrylic cylinder (60 × 30 cm, high and diameter, respectively) filled with water (40 cm) at a temperature of 23 ± 1 °C. Animals were singly placed in the center of apparatus and free exploitation was allowed for 300 s. The first 120 s were admitted as habituation stage. Immobility time (floating with minimal movements) and climbing (upward-type activity) were recorded by an experimenter in the last 3 min of the test [40,41]. The augment of immobility time indicates depressive-like phenotype.

### 2.4. Oxidative Biochemistry Assays

Two hours after behavioral assays, animals (*n* = 3–5 animals per group) were anesthetized with isoflurane (2%), euthanized by cervical dislocation to collect prefrontal cortex and hippocampus tissues for analysis. Biological samples were submitted to liquid nitrogen and subsequently stored in freezer at −80 °C until analysis. To analyze the samples, the tissues were thawed and resuspended in TrisHCl (20 mM, pH 7.4, at 4 °C), through sonic disaggregation (approximate concentration of 1 g tissue/mL). Biochemical analyzes of Antioxidant Capacity Against Radicals Peroxyl (ACAP), reduced glutathione (GSH), lipid peroxidation (LPO), protein content, and reactive oxygen species (ROS) were assessed. ACAP evaluation was chosen to characterize the antioxidant non-enzymatic status, which is strictly correlated to GSH content [42]. ROS levels demonstrate the unbalance of antioxidant defenses versus the overproduction of prooxidant factors. In addition, LPO was performed to reflect the oxidative damage by cell lipid peroxidation mechanism [43]. All biochemical analyses were normalized by protein content levels. Data were shown as percentual of control group.

#### 2.4.1. Antioxidant Capacity against Radicals Peroxyl (ACAP)

ACAP analysis was carried out through the protocol adapted from Amado et al. [44]. ACAP levels were determined by the amounts of ROS activated by treatment with peroxyl radicals. Lysed tissues were prepared in a 1:5 weight/volume ratio associated with a buffer solution of 100 mM Tris-HCl (pH 7.75) plus EDTA (2 mM) and Mg^2+^ (5 mM). The homogenate was centrifuged at 10,000 rpm for 20 min at 4 °C. In 96-well microplates, 10 microliters of supernatant from each tissue were applied and protein concentrations were pipetted into 6 wells per sample. Reactive buffer (127 microliters) containing 30 mM zwitterionic sulfonic acid [HEPES] (pH 7.2), 200 mM KCl, and 1 mM MgCl_2_ was added to the samples. In three of five samples, 7.5 µL of 2,2′-azobis 2 methylpropionamidine dihydrochloride (ABAP; 4 mM; Sigma-Aldrich) were pipetted. The other two samples were pipetted with the same volume of pure water. In the sequence, we added 10 µL of the fluorescent probe (2′,7′, dichlorofluorescein diacetate-H2DCF-DA) with a final concentration of 40 micromoles [44] to all wells. The lecture was performed by spectrophotometry at wavelengths of 488 nm and 525 nm, and temperature of 35 °C. At this temperature, peroxyl radicals are produced by thermal decomposition of ABAP. The non-fluorescent component H2DCF is oxidized by ROS and the component fluorescent H2DCF was detected by excitation and emission, respectively. ABAP decomposition and formation of ROS were monitored for 30 min with intervals of readings every 5 min [42].

#### 2.4.2. Reduced Glutathione (GSH)

GSH intracellular levels assessment is based on the ability of GSH to reduce 5,5-dithiobis-2-nitrobenzoic acid (DTNB) molecules (Sigma-Aldrich) to nitrobenzoic acid (TNB), which can be measured by spectrophotometry (wavelength of 412 nm) [43]. Briefly, an aliquot of 20 μL of homogenized tissue was mixed with 20 μL of distilled water plus 3 mL of PBS/EDTA, and posteriorly performed the first reading (T0). Then, 100 μL of DTNB was added to the mixture for the second measurement (T3) after 3 min.

#### 2.4.3. Lipid Peroxidation (LPO) Levels

To evaluate LPO, the MDA present in the tissue samples (produced by LPO) interacts with thiobarbituric acid (TBA) forming MDA–TBA reaction products. Briefly, 100 µL of tissue samples was mixed with 900 µL of water (main solution) to collect an aliquot of 0.5 mL to add in a tube containing TBA compound (1 mL; 10 nM). This mixture was submitted to 60 min of water bath (94 °C), following the addition of 4 mL of n-butyl alcohol (Sigma-Aldrich), stirring in vortex device and centrifugation (175× *g*) for 15 min. The lecture of 3 mL of the supernatant was employed by spectrophotometry (535 nm) [45].

#### 2.4.4. Protein Content

Initially, brain tissue samples were weighed and homogenized in cryogenic mill (Freezer Mill 6770—Spex Sample Preparation, Metuchen, NJ, USA), and cryofractured at low temperatures (liquid nitrogen). Then, tissue samples (50 μg) were added to 500 μL of lysis buffer A (Urea/Thiourea), incubated for 2:30 h in refrigerator with constant agitation, following centrifugation at 14,000 rpm for 30 min at 4 °C. Supernatant was collected and submitted to Bradford assay for total protein quantification, using the Quick Start TM Bradford Protein Assay kit (Bio-rad, Hercules, CA, USA), in duplicate, as described previously [45]. For each reaction, 5 μL of sample plus 250 μL of staining reagent was used. After incubation for 5 min at room temperature, absorbances were determined at 595 nm in spectrophotometer. Protein concentrations were calculated by comparing with a standard curve containing Bradford concentrations.

#### 2.4.5. Reactive Oxygen Species (ROS)

The prefrontal cortex and hippocampus (*n* = 5 animals/group) were collected, frozen in cryoprotection liquid (Tissue-Tek), and submitted to cryostat equipment (10 µm section). Tissue sections were incubated with the dihydroethidium (DHE) probe for 30 min (concentration of 2.5 × 10^−6^ M) in a dark and humid chamber. Then, the sections were washed in PBS to be photographed in a Fluorescence Microscope (Zeiss). Each animal slide was used to capture two different fields of the prefrontal cortex and hippocampus. Finally, the images were opened in the Image J software to quantify ROS content. A rectangle (60 w × 60 h) was drawn using the rectangle tool option to measure the fluorescence intensity in 30 different regions containing a labeled nucleus. The average of the two measured photos was calculated to obtain the media value of the animal sample [46,47].

### 2.5. Hippocampal Brain-Derived Neurotrophic Factor (BDNF) Levels

Because the hippocampus consists of a primary brain region related to neurogenesis that plays a pivotal role in emotionality, we assessed BDNF levels in hippocampal slices through reverse transcription qPCR (RT-qPCR) technique. For this, in the euthanizing stage of experiments, hippocampus was dissected (*n* = 5 animals/group), placed in tubes containing RNA later and stored at −80 °C.

In accordance with the manufacturer’s guideline for cDNA synthesis, we used GoScript™ Reverse Transcription System (Promega Corporation, Madison, WI, USA, EUA). To measure BDNF levels by real-time PCR (qPCR), the GoTaq^®^ Probe qPCR Master Mix (Promega Corp.) was employed, as reported previously [48]. We performed all assays in triplicate, employing 96-well PCR plates, in the CFX96 Touch™ Real-Time PCR Detection System equipment (Bio-Rad, USA). The final BDNF expression evaluation was obtained through the Bio-Rad CFX Manager™ 3.1 software (Bio-Rad) and normalized by Actb in control samples. The MIQE guidelines were followed, and the relative gene expression was assessed through the formula 2−ΔΔCT (*p* < 0.05). Bdnf and Actb expression was measured by Taqman^®^ gene expression assays (Applied Biosystems, Waltham, MA, USA) (Rn02531967 and Rn00667869, respectively).

### 2.6. Statistical Analysis

For statistical evaluation, Gaussian distribution was performed by Kolmogorov–Smirnov test for behavioral analysis. Behavioral statistical comparisons between the groups were performed using Student’s *t*-test. In biological samples, we applied the non-parametric test of Mann–Whitney evaluation. Results were expressed as mean ± standard error of the mean (SEM). Values of *p* < 0.05 were considered statistically significant.

## 3. Results

### 3.1. Intermittent Ketamine Exposure during Adolescence Reduces Total and Central Spontaneous Exploratory Behavior in the Early Withdrawal Stage in Female Rats

A cycle of three daily consecutive administrations of a recreational dose of ketamine (10 mg/kg) during early adolescence reduces spontaneous horizontal exploratory ambulation (9.612 ± 1.261; *p* < 0.0001) but not vertical (0.3636 ± 0.5270; *p* > 0.05) in the early withdrawal (24 h) stage on open field apparatus (Figure 2A,B). In addition, ketamine exposure reduced the central distance traveled (−2.233 ± 0.3787; *p* < 0.0001), as well as the time spent in the center of the apparatus (−10.92 ± 2.195; *p* < 0.0001; Figure 2C,D).

### 3.2. Intermittent Ketamine Exposure during Adolescence Induces Psychiatric-like Phenotype Related to Anxiety and Depression in the Early Withdrawal Stage in Female Rats

One cycle of intermittent administration of a recreational dose of ketamine during adolescence induces an anxiogenic-type profile in an elevated plus-maze test (Figure 3). In accordance with open field findings, the percentual of open arms time was reduced (−8.273 ± 2.472; *p* < 0.01) but not open arms entries (−0.09091 ± 3.082; *p* = 0.9768), which suggests anxiogenic-like behavior (Figure 3A,B). The enclosed arms entries parameter was not modified (Figure 3C).

In addition, to evaluate depressive-like features, we employed the forced swimming test. Our data show that ketamine exposure subjects exhibited an augment of immobility time (38.64 ± 8.798; *p* < 0.0003), which indicates a depressive-like phenotype (Figure 4). In addition, climbing time, which was adopted as a motor parameter, was not modified (−12.64 ± 7.147; Figure 4B).

### 3.3. Emotionality Alteration Induced by Intermittent Ketamine Exposure during Adolescence Was Associated to Overproduction of ROS and Evidence of Oxidative Damage on Prefrontal Cortex and Hippocampus in the Early Withdrawal Stage in Female Rats

The Mann–Whitney test revealed that one cycle of intermittent ketamine exposure in adolescent female rats altered the oxidative biochemistry with evidence of oxidative stress on the prefrontal cortex and hippocampus in the early withdrawal stage (Figure 5). In fact, antioxidant parameters (i.e., ACAP and GSH) were negatively altered in a brain region-dependent manner. The hippocampus exhibited ACAP and GSH level reductions (ACAP: 96.99 ± 0.3454%; *p* < 0.05; GSH: 43.63 ± 1.155%; *p* < 0.01; Figure 5A,B); whereas the prefrontal cortex presented only ACAP reduction (95.68 ± 0.4768%; Figure 5D). In addition, LPO was found in all brain areas evaluated (hippocampus: 99.46 ± 0.1543; *p* = 0.0228; prefrontal cortex: 101.5 ± 0.3729; *p* = 0.0017; Figure 5C,F).

Therefore, one cycle of intermittent ketamine exposure elicited overproduction of ROS in both brain areas, i.e., the prefrontal cortex (162.1 ± 11.59%; *p* < 0.001; Figure 6A–C) and hippocampus (124.7 ± 5.239%; *p* < 0.01; Figure 6D–F). Photomicrography related to the median sample shows the increased tissue labeling in ketamine-exposed animals (Figure 6B,E).

All these findings suggest that oxidative damage occurs in the ketamine exposure group in the early withdrawal stage in female rats.

### 3.4. Hippocampal BDNF Levels Were Not Altered in the Early Withdrawal Stage of One Cycle of Ketamine Exposure in Adolescent Female Rats

One cycle of recreational dose of ketamine administration did not modify BDNF levels in the hippocampus of adolescent female rats in the early withdrawal stage (ketamine: 97.02 ± 15.32%; *p* = 0.8140; Figure 7).

## 4. Discussion

As a result of its hallucinogenic and dissociative effects, ketamine has grown in recreational environments as an illicit drug [4,49]. Hallucinogenic effects were described for the first time during voluntary trials, particularly in clinical use [19]. However, the interest in the recreational use of ketamine has expanded in recent years, and illicit consumption has increased due to its easy availability and low price. This fact has raised new health concerns, mainly related to physically and psychologically harmful consequences [50]. In this context, the present study revealed that recreational exposure to ketamine elicited negative behavioral effects related to anxiety and depression profiles in early withdrawal. Such behavioral disorders were associated with ROS overproduction and evidence of oxidative damage.

The use of recreational drugs is harmful at all stages of life; however, it is critical during neurodevelopment, as during the adolescence period, fundamental changes in brain structures occur [51,52]. The adolescent brain undergoes an intense maturation process that attributes higher vulnerabilities to psychotropic substance use [51]. According to Spear [29], adolescence is the transition from childhood to adulthood [29], in which modifications in physical, psychosocial, and cognitive characteristics are detected [52]. In humans, adolescence corresponds to 10 to 20 years of age [53,54], which in rodents comprises the period from 28 to 42 postnatal days [24,29].

In fact, a variety of substances of abuse consumed in “club drugs” (i.e., clubs, raves, and bars) exist, which are frequently used among teenagers and young adults with harmful effects on the CNS [55]. Among these recreational drugs, ketamine emerges as an NPS, therapeutically used as a dissociative anesthetic in humans and veterinary medicine [56]. Typical recreational doses of ketamine used can vary from 10 to 300 mg/day, depending on the route of administration [15,57]. Despite studies demonstrating the hallucinogenic effects of ketamine, in particular in an illicit context, the literature is scarce when describing the behavioral effects of this substance during the period of abstinence [15]. Even considering the translational nature related to experimental and clinical investigations, our group employed a pattern of recreational use related to occasional users (i.e., non-dependent consumers), reflecting one weekend of drug consumption, as occurs in the atmosphere of clubs, bars or raves, involving adolescent users aged 12 to 25 years of age [56]. Thus, the present study aimed to mimic a scenario of exposure to ketamine in a recreational paradigm (i.e., one cycle of the subanesthetic dose of 10 mg/kg/day), to characterize the neurobehavioral disorders and oxidative consequences in the early abstinence stage in female rats.

The present results revealed negative behavioral disorders related to a reduction in spontaneous ambulation, as well as anxiogenic- and depressive-like phenotypes. In the open field test, the total distance traveled (horizontal exploration) reflects the natural exploratory ambulation of the animal, which requires motivational inputs [39,57]. Ketamine subanesthetic dose administration elicits hyperlocomotion, which characterizes stimulant drugs [58,59]. The blockade of NMDA receptors, located in interneurons of the striatum, nucleus accumbens, and prefrontal cortex regions, results in an increase in glutamate and dopamine neurotransmitter availability, and consequently hyperlocomotion [60,61]. However, in the immediate abstinence (i.e., three hours after the ketamine challenge protocol), hyperlocomotion was no longer found [28]. Of interest, our findings reveal that in the early withdrawal period, twenty-four hours after subanesthetic administration of ketamine, horizontal exploratory activity was impaired. Despite the lack of quantification of dopamine in the critical brain areas mentioned above, we suggest that early withdrawal from ketamine elicits disruption of neurotransmitter system balance, such as dopamine, noradrenaline, and serotonin, which might alter behavioral tests that require motivational skills, such as spontaneous ambulation [62]. Thus, this scientific evidence suggests that ketamine has several abilities to influence the parameters of the open-field test, due to the properties of NMDA-type glutamate receptor blockade and dopaminergic modulation [63].

In order to encompass the study of emotional responses, distance and time in the center of open field apparatus were measured [35]. We found that in the early withdrawal period, individuals that received one cycle of subanesthetic dose of ketamine reduced both arena center exploitations (i.e., central distance traveled and central time parameters). Consistent with these initial findings, ketamine 24 h-withdrawal animals reduced the percentual of open arms time in the elevated plus-maze paradigm, which consists of a more predictive indicator to analyze anxiety-like behavior in rodents [64]. All these data suggest that early ketamine withdrawal is characterized by an anxiety-like phenotype in adolescent female rats.

In addition to anxiogenic-like behavior, early ketamine withdrawal displayed a depressive-like profile in the forced swimming test challenge, assessed by the increment of immobility time [41]. Notably, anxiety and depression consist of the most reported problems by misusers seeking drug addiction treatment [60]. In general, there is a close relationship between depression and anxiety co-morbidities, regardless of the type of recreational drug used [65]. Anxiety usually occurs during the acute phase of withdrawal from some drugs, such as alcohol, and can persist for up to 2 years as part of post-acute abstinence in about a quarter of recovering people [66]. On the other hand, depression was found to be more critically related to drug withdrawal, as a result of the high prevalence and the negative consequences that are generated [67].

Regarding ketamine use, the first proposal of abstinence effects in a clinical study was carried out by Curran and Morgan [6]. These authors selected 39 young volunteers (23 men and 16 women), aged 18 to 29 years old. The dose of ketamine used was approximately 2 mg/kg, which is greater than the 0.1–1.0 mg/kg used in laboratory research, but similar to doses reported as being used by recreational users [6]. In this study, it was observed that ketamine causes working, episodic, and semantic memory impairments, as well as psychogenic and dissociative effects, which may reflect the chronic or residual effects of drug use [6]. Subsequently, case report studies reported that after the first four hours of ketamine withdrawal, patients experienced memory loss, psychotomimetic symptoms, and anxious behavior [68,69]. In line with this, depressive behavior in ketamine users was reported. The randomized double-blind study evaluated 150 subjects, subdivided into frequent ketamine users, infrequent ketamine users, abstinent misusers, polychrome controls, and non-users of illicit drugs. The authors found an increase in depression scores over the 12 months of the study in the frequent user and ketamine withdrawal groups [69]. In addition, Morgan and colleagues [19] reported that depression disorder was more frequent in ketamine withdrawal users, reflected in higher events of negative experiences over the 12 months of the study than in other groups [19]. Actually, several clinical investigations have revealed that among the misusers under ketamine addiction treatment, about half of these patients experience long-term depressive and anxiety symptoms, among other psychiatric conditions [70,71]. Despite these reports, Morgan and Curran (2011) claim that there is conflicting evidence about the exact characteristics of early withdrawal after cessation of ketamine use [19].

In summary, anxiety and depression are common psychiatric disorders in ketamine abusers’ withdrawal. Accordingly, an experimental study conducted by our group revealed similar responses to emotionality in female Wistar rats after the acute administration of a recreational dose of ketamine. In addition, behavioral alterations were associated with peripherical (plasma) and central (hippocampus) oxidative stress [28].

In fact, there is a robust relationship between anxiety behavior and oxidative stress, but the influence of other brain systems cannot be neglected [72]. Oxidative balance on the CNS or plasma is an important factor for behavioral alterations, such as anxiety and depression [73,74]. Our findings also found that twenty-four-hour abstinence from ketamine resulted in evidence of oxidative damage, parallel to reduced ACAP content and increased levels of ROS on the prefrontal cortex and hippocampus, as well as reduced GSH levels that represent one of the “first line” of protection against oxidative stress on the hippocampus of adolescent female rats. Likewise, ACAP allows for characterizing the general antioxidant status through peroxyl radical evaluation, which involves different components of enzymatic and non-enzymatic antioxidants [74]. Therefore, the reduction of the antioxidant capacity negatively affects the antioxidant components to minimize the free radical overproduction, compromising cell function [75].

Therefore, ACAP determination may provide a better understanding of how antioxidants interact with ROS [76]. The advantage of this simple technique is the ability to establish an integrated antioxidant response against ROS, of which antioxidant alterations may be a consequence of a damaged oxidative state [77].

GSH, the most abundant non-enzymatic antioxidant that acts by eliminating hydroxyl radicals in cells [78], consists of a robust defense against ROS, and participates in numerous cellular reactions, neutralizing pro-oxidants molecules or catalyzing enzymatic reactions [79]. In the present study, ketamine exposure during adolescence reduced the levels of GSH in the hippocampus. This finding suggests that GSH plays a fundamental role in cellular defense and detoxification in CNS structures with an impact on behavioral domains [79].

For Zhang et al. [80], reduced GSH levels were associated with depressive-like disorders in a chronic pain rat model [78]. Indeed, scientific evidence demonstrated the occurrence of high levels of ROS in depressive patients [81,82]. Experimental studies have reported the involvement of oxidative unbalance as an important pathway in the pathophysiology of mood disorders [81]. Notably, preclinical studies suggest that the hippocampus, amygdala, and prefrontal cortex are more vulnerable to oxidative stress than other brain structures, which may elicit behavioral impairment [83,84,85,86,87].

ROS represents a series of oxygen molecules [i.e., superoxide (O2^•−^)] or oxygen-containing free radicals [i.e., hydrogen (H_2_O_2_)], which is generated normally during cell metabolism [82,87]. Thus, ROS is recognized for playing a double role, (i) beneficial effects in the defense against infectious agents, body systems signaling, or induction of mitogenic response; (ii) and deleterious effects in the occurrence of oxidative stress [88]. This latter consequence occurs due to the elevated degree of free radical reactivity, in which the overproduction of these reactive species generates biological damage, namely oxidative stress [83].

In the present study, ketamine exposure displayed an augmentation of ROS on the prefrontal cortex and hippocampus of adolescent rats during early withdrawal. According to Tsikas et al. [78], these events are consequences of the high reactivity of ROS, which generates successive attacks on several local biomolecules, including proteins, DNA, and lipids, such as polyunsaturated fatty acids (PUFAs) [89]. Lipid intermittent “aggressions” generate the formation of MDA, 4-hydroxy-2-nonenal (HNE) and other reaction products, such as F2-isoprostanes [78], also called LPO [89,90]. LPO products of polyunsaturated fatty acids and their esters exert deleterious effects, such as cytotoxicity and modification of proteins and DNA bases [91,92]. Therefore, extensive LPO alters membrane assembly, composition, structure, and function during all periods of life [91]. In fact, reduced gray matter volume and white matter development consist of a critical factor that contributes to the prefrontal cortex’s greater vulnerability during adolescence [92]. At this stage of life, dendritic restructuring (called “pruning”) also occurs, which allows the eradication of synapses, facilitating the effectivity of synapse connections in specific brain regions [93].

According to Bouayed et al. [74], there is a close relationship between oxidative stress and psychiatric disorders, including depression, anxiety, and schizophrenia [74,94].

An experimental study, using acute recreational doses of ketamine challenge, reported an increase in LPO levels in several brain areas, such as the cerebellum, the cortex, the striatum, the prefrontal cortex, and the hippocampus, in a region- and dose-dependent manner in male adult rats [93,94]. In fact, such a study demonstrated that upon immediate abstinence (i.e., 3 h), oxidative stress was not counterbalanced and was still detected in the hippocampus and plasma [28]. Based on the present data, our theory relies on the suggestion that this oxidative damage on brain structures persists in the early withdrawal period (24h of abstinence). In consonance with this hypothesis, we speculate that such an undesirable oxidative profile may be aggravating the immature brain, as with adolescent subjects.

Actually, the prefrontal cortex dorsolateral presents brain association areas that integrate high levels of information from numerous sensory modalities, which play a vital role in executive functions, such as planning, organization, and motor regulation [95]. This brain area reaches maturity between 16–17 years of age in humans, which confers higher vulnerability on the prefrontal cortex to psychotropic agent exposure [95,96].

In fact, the prefrontal cortex refinement and maturation processes provide dendritic pruning to trigger the synaptic efficiency of neural circuitry, that in turn modifies gray and white matter volume [92]. On the other hand, the hippocampus represents a vital area of the brain that mediates complex emotional (i.e., anxiety and depression) and cognitive functions, which undergo extensive connectivity and refinement modifications during adolescence [92,93]. To support such complex circuitry, hippocampal regions present a high density of specific receptors related to stressors stimuli, which confers peculiar vulnerability to neurochemical environmental alterations, affecting mood and cognitive domains [87].

Several studies have suggested that CA1, CA3 pyramidal cells, and dentate gyrus of the hippocampus are susceptible to hazardous effects of stressor repercussions (i.e., oxidative damage), displaying functional negative outcomes in learning and memory, anxiety, and depression [97,98]. Albeit ketamine improved GSH and reduced MDA levels in a chronic pain adult rat model, 8 days after acute single ketamine injection, our findings indicated that 3 subsequent days of recreational doses of ketamine during adolescence reduced GSH levels on the hippocampus, as well as MDA levels in the prefrontal cortex and hippocampus, which are structures closely related to depressive-like features [74].

It is noteworthy that BDNF expression was associated with the neurotrophic theory of depression etiology [99]. The current work fails to prove that the neurotrophic factor contributed to the emotional disruption noted in the ketamine-abstinent group since hippocampal BDNF levels were not modified in the 24 h ketamine-abstinence subjects. Experimental research reported that an acute dose of ketamine increases hippocampal BDNF expression in depressant-like adult rats induced by chronic postsurgical pain, mitigating depression-like features [100]. We claim that these controversial findings are a result of (i) the stressful model, since the authors did not measure the ketamine effects on surgical-induced subjects against control subjects; (ii) the mature brain of adult animals tested; and (iii) the regimen of a single acute dose of ketamine [100]. We elucubrate that the recreational use of ketamine returns to normal levels during 24 h of withdrawal in adolescent female rats, which reduces its neuroprotection related to behavioral disorders.

Finally, we highlight the limitation of the preclinical studies. Of course, due to the reduced number of samples, restricted methodologies applied, and inherent characteristics between species, the translational nature of the present findings is restricted. However, our study aims to characterize as well as highlight interesting possibilities for further studies but not conclude the complex state-of-the-art information related to ketamine recreational use during adolescence.

## 5. Conclusions

In conclusion, one cycle of 3 consecutive days of ketamine elicits spontaneous motor activity reduction and emotional (anxiety- and depressive-like) behavioral alteration in the early withdrawal period. In addition, evidence of oxidative damage pathways may contribute to behavioral disorders found but not BDNF expression. Thus, these data demonstrate that recreational ketamine use during adolescence disrupts vital behavioral brain structures with consequent changes in emotionality in adolescent female rats.

## Figures and Tables

**Figure 1 pharmaceuticals-15-01373-f001:**
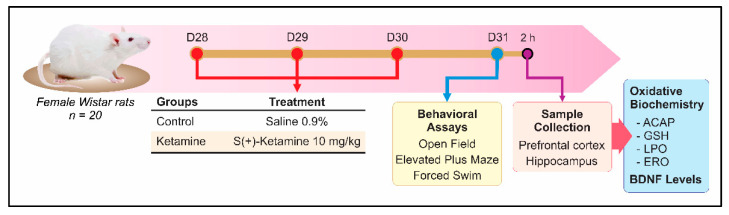
Experimental study design. Adolescent rats (28 to 30 days old) received saline or ketamine (10 mg/kg/day) intraperitoneally for 3 consecutive days. Twenty-four hours after the last administration, animals were sequentially submitted to open field, elevated plus maze, and forced swimming behavioral tests. Two hours after the behavioral tests, animals were anesthetized and euthanized for biological sample collection for biochemical assays.

**Figure 2 pharmaceuticals-15-01373-f002:**
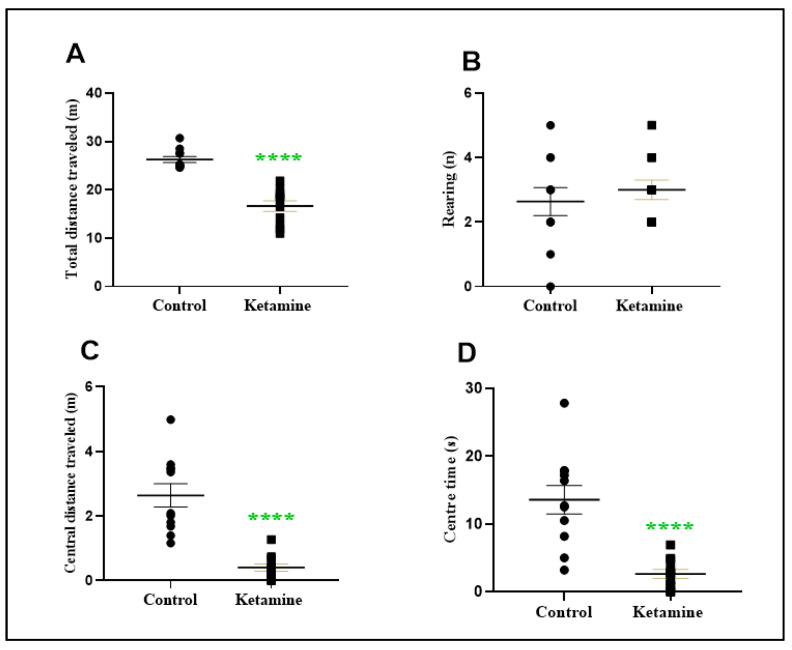
Effects of ketamine recreational regimen (10 mg/kg/day for 3 days, i.p) or saline (control group) in the early withdrawal (24 h) stage on open field test of adolescent female Wistar rats (*n* = 10 animals/group). Panel (**A**) represents the total distance traveled; Panel (**B**) represents the vertical exploitation in absolute numbers; Panel (**C**) represents the central distance traveled in meters; Panel (**D**) represents the time spent in central area. Results were expressed as mean ± S.E.M. **** *p* < 0.0001 compared to control group. Student *t*-test.

**Figure 3 pharmaceuticals-15-01373-f003:**
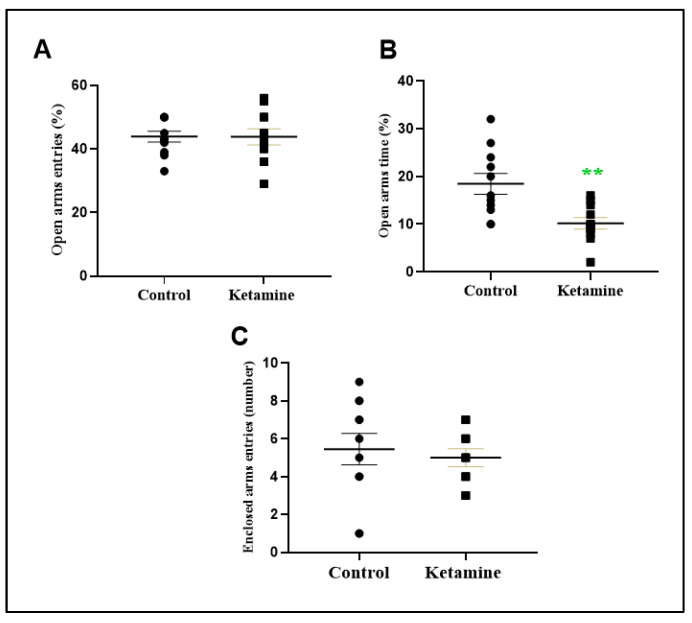
Effects of ketamine recreational regimen (10 mg/kg/day for 3 days, i.p) or saline (control group) in the early withdrawal (24 h) stage on anxiety-like parameters in elevated plus-maze test. Panel (**A**) represents the percentual of open arms entries; Panel (**B**) represents the percentual of open arms time; Panel (**C**) represents the entries of enclosed arms (number) of adolescent female Wistar rats (*n* = 10 animals/group). Results were expressed as mean ± S.E.M. ** *p* < 0.01 compared to control group. Student *t*-test.

**Figure 4 pharmaceuticals-15-01373-f004:**
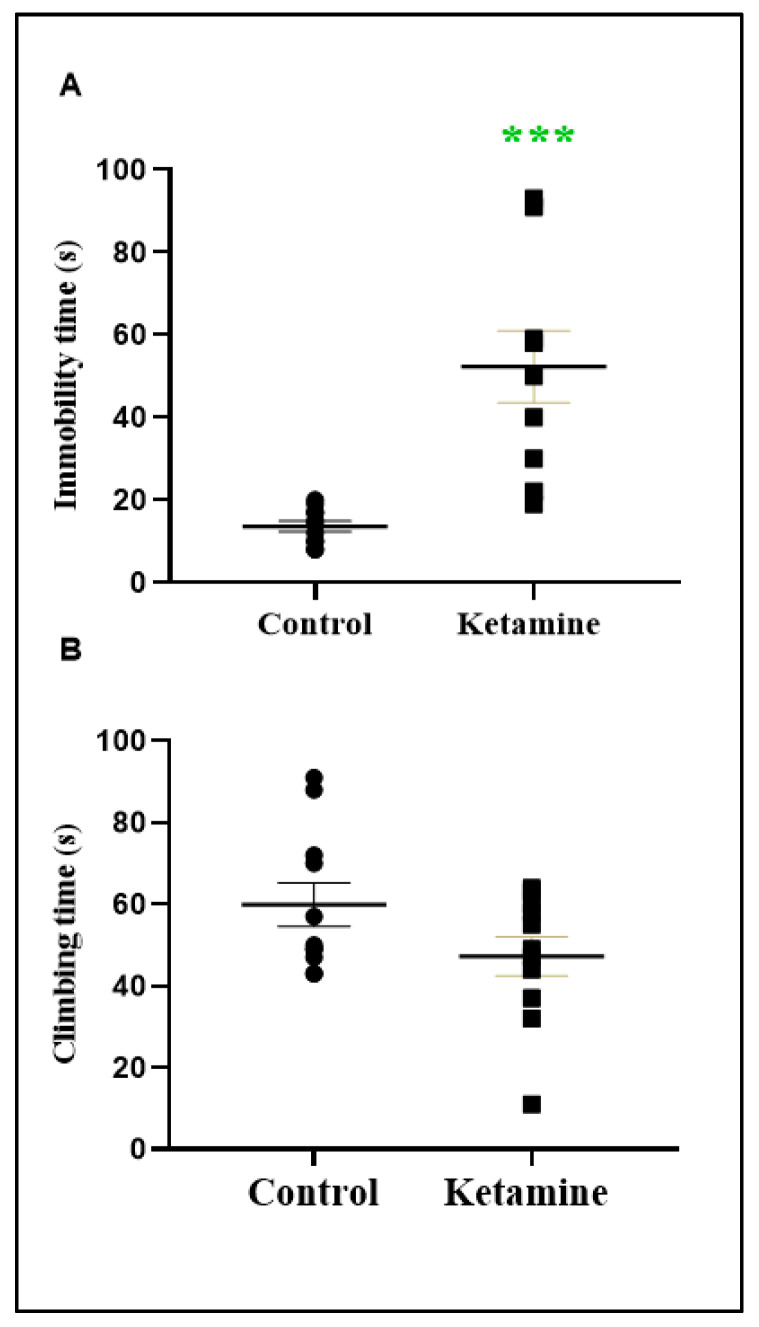
Effects of ketamine recreational regimen (10 mg/kg/day for 3 days, i.p) or saline (control group) in the early withdrawal (24 h) stage on depressive-like parameters in forced swim test of adolescent female Wistar rats (*n* = 10 animals/group). Panel (**A**) represents the immobility time (seconds); Panel (**B**) represents the climbing time (seconds). Results were expressed as mean ± S.E.M. *** *p* < 0.001 compared to control group. Student *t*-test.

**Figure 5 pharmaceuticals-15-01373-f005:**
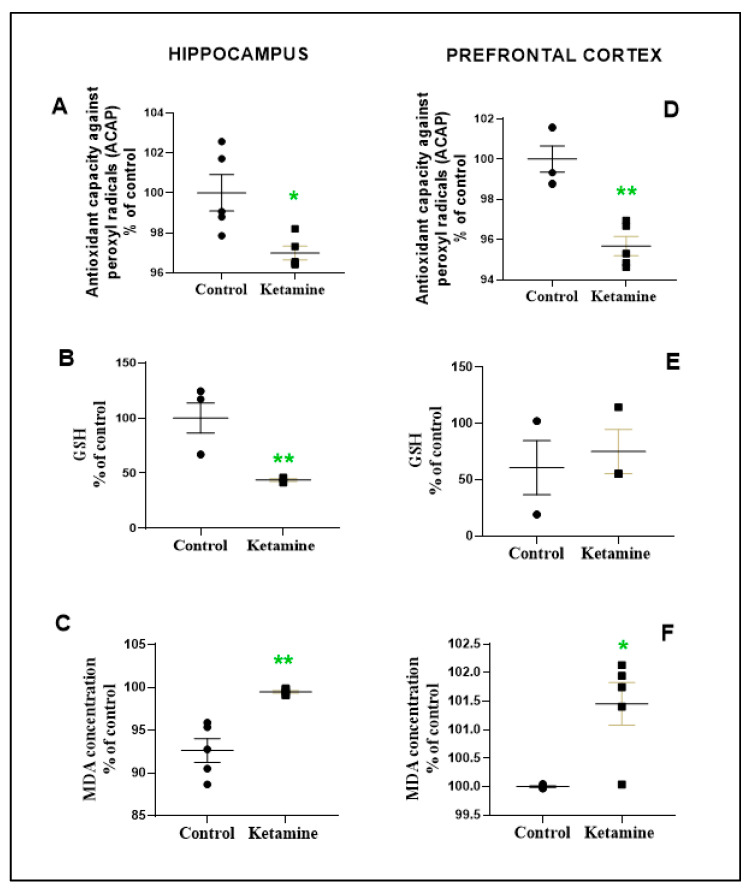
Effects of ketamine recreational regimen (10 mg/kg/day for 3 days, i.p) or saline (control group) in the early withdrawal (24 h) stage on oxidative balance parameters on hippocampus (**A**–**C**) and prefrontal cortex (**D**–**F**) of adolescent female Wistar rats (*n* = 3–5 animals/group). Panel (**A**,**D**): Antioxidant Capacity Against Radicals Peroxyl (ACAP); panel B and E: Reduced glutathione (GSH); panel C and F: malondialdehyde (MDA). Results were normalized by protein content and expressed as percentual of control group. * *p* < 0.05 compared to control group; ** *p* < 0.01 compared to control group. Mann–Whitney test.

**Figure 6 pharmaceuticals-15-01373-f006:**
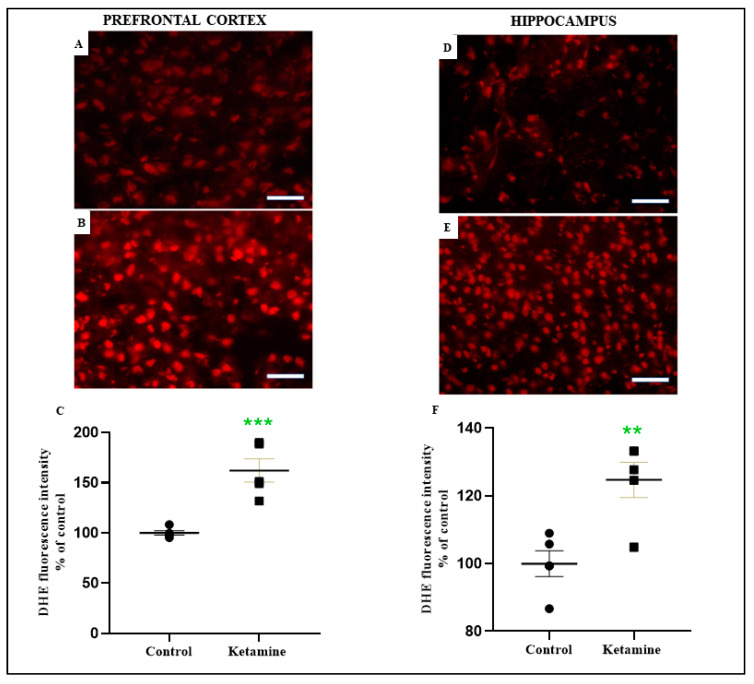
Effects of ketamine recreational regimen (10 mg/kg/day for 3 days, i.p) or saline (control group) in the early withdrawal (24 h) on reactive species oxygen (ROS) levels on prefrontal cortex and hippocampus of adolescent female Wistar rats (*n* = 5 animals/group). i. Representative fluorescence microscope photomicrographs of prefrontal cortex (**A**,**B**) and hippocampus (**D**,**E**) sections, incubated with dihydroethidium (DHE) probe. Red fluorescence reflects ROS production. ii. Quantification of DHE fluorescence intensity on prefrontal cortex (**C**). iii. Quantification of DHE fluorescence intensity on hippocampus (**F**). Results were expressed as percentual of the control group. *** *p* < 0.01 compared to control group; ** *p* < 0.0001 compared to control group. Mann–Whitney test. Scale bar: 50 µm.

**Figure 7 pharmaceuticals-15-01373-f007:**
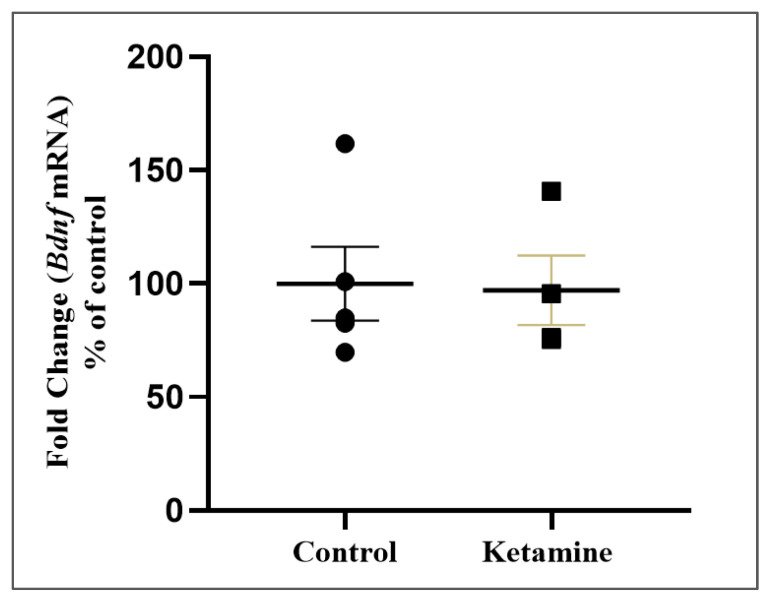
Effects of ketamine recreational regimen (10 mg/kg/day for 3 days, i.p) or saline (control group) in the early withdrawal stage (24 h) on hippocampal brain-derived neurotrophic factor (BDNF) levels of adolescent female Wistar rats (*n* = 5 animals/group). Results were expressed as percentual of control group. Mann–Whitney test.

## Data Availability

Data is contained within the article.

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
