# Peer review of "“K-Powder” Exposure during Adolescence Elicits Psychiatric Disturbances Associated with Oxidative Stress in Female Rats"

_pharmaceuticals, 2022, doi:10.3390/ph15111373_

Round 1

Reviewer 1 Report

This animal study aims to explain the behavioral and neurobiological consequences of short-term recreational use of ketamine during a particularly vulnerable period (adolescence). This is certainly relevant, but the manuscript needs extensive revision. It lacks a clear motivation, and the conclusions or parts of them are based on a post-mortem analysis that are underpowered.  It also needs fundamental restructuring.  See further clarifications below:

1.       No definition of K-powder given. Why not just use of Ketamine in the title

2.       Please provide information on the employed Ketamin, Supplier, number…. (mixture dextro /leva or not and the known differences between the two and relevant implication for psycoactivity, https://www.ncbi.nlm.nih.gov/pmc/articles/PMC5490113/

3.       There is zero information on ketamine and oxidative stress within introduction, which would somehow motivate the analysis thereof.

4.       Within the introduction please add key findings from ref 25, which seem crucial in understanding the motivation behind the study and the differences towards this study

5.       For all figures individual datapoints or alternatively boxplots/violin plots need to be added/used to provide information of data distribution in graphs

6.       Design and experimental timeline are clear. However, it needs to be clarified, how many animals and why (difference in sample preparations?), from the behavioral group where assigned to which post-mortem analysis

7.       Testing of normal distribution is non-sense for all post-mortem analysis if indeed data comes from 3-5 animals. As the sample size is just to small to even test for normal distribution. Thus, it is questioned if testing as described under 2.6 by KS-Method was really performed or if its interpretation is unclear to the authors. However, either way t-test is not appropriate and should be replaced with non-parametric testing.  Finally, it should be acknowledged that the sample sizes are very small, assumingly no power calculations have been performed before the experiment was set up.

8.       Please explain your choice of post-mortem tests for each of the analyzed outcomes. What is e.g. TEAC, what does it mean, what does it indicate. For all post-mortem assessments the method used is explained sufficiently but the rational is missing.

9.   In the same direction it is questionable why so many different post-mortem markers were analyzed as this results in the limitations described above. While, likely not applying to behavioral data, postmortem analyses strike me as underpowered. It may have been more favorable to restrict analysis to a number of outcomes taken from all animals of the behavioral group. Taken that all suggestions will be carefully considered and addressed all results and assumptions based on the presented post-mortem data must then still be regarded as preliminary. In addition, a limitation section needs to be included. Please also consider this for the choice of your titel.

10.   Results Part Behavior: I do not appreciate the split of outcome variables from on behavioral test in different paragraphs and more importantly figures. This makes the result unnecessary complicated to understand. For example, in Figure 2 it takes to much effort to find out if A responds to open-field or EPM.  For the results part one paragraph summarizing the behavioural findings will be sufficient.

11.   For ERO how many sections/ROIs from how many animals were counted. How was the ROI defined, ROI size …., what do you mean by quantified ROI measured through Image J (potentially English mistake?). In addition, please add detailed information on analyzed ROI for example providing an overview image. Which hippocampus subregion/s C1,C2,C3?/ dorsal/ventral were analyzed. Photomicrographs also miss scale bars. The panel information in Figure legend is hard to understand, either label A) control hippocampus, B) …. , or indicate with labels directly within figure. Also changing terminology between ERO and ROS is confusing.

12.   Discussion needs substantial rewriting: overall to lengthy and some of the information should be considered to be moved to the introduction. For example it takes 3 paragraphs to end with “ Thus, the present study aimed to mimic a scenario of exposure to ketamine in a recreational paradigm (i.e., one cycle of the subanesthetic dose of 10mg/kg/day) to characterize the neurobehavioral disorders and oxidative consequences in the early abstinence in female rats.” Which represents the Motivation of the study. You either shorten this substantially or you start with this sentence and move everything before to the introduction.

13.   Conclusion “behavioral disorder” should be behavioral alteration or phenotype

14.   English proofreading required

Reviewer 2 Report

See attached.

Round 2

Reviewer 1 Report

I appreciate the careful revision of the manuscript. The majority of concerns were appropriately addressed. 

Regarding point 5.) Not all figures (bar graphs) were changed to provide individual data points.

In addition, please check the correction to Mann-Whitney test throughout the manuscript e.g. Paragraph 3.3 starts with „Stutent t test…“ which needs to be changed.
